# The Role of Ultrasound in Cancer and Cancer-Related Pain—A Bibliometric Analysis and Future Perspectives

**DOI:** 10.3390/s23167290

**Published:** 2023-08-21

**Authors:** Badrinathan Sridharan, Alok Kumar Sharma, Hae Gyun Lim

**Affiliations:** 1Department of Biomedical Engineering, Pukyong National University, Busan 48513, Republic of Korea; badri@pknu.ac.kr; 2Department of Information Management, Chaoyang University of Technology, Taichung 413310, Taiwan; rbaloksharma@gmail.com

**Keywords:** acousitc waves, ultrasound, bibliometry, tumor, pain, nociception

## Abstract

Ultrasound has a deep penetrating ability with minimal or no tissue injury, while cancer-mediated complications during diagnosis, therapy, and surgery have become a serious challenge for clinicians and lead to the severity of the primary condition (cancer). The current study highlights the importance of ultrasound imaging and focused ultrasound therapy during cancer diagnosis, pain reduction, guidance for surgical resection of cancer, and the effectiveness of chemotherapy. We performed the bibliometric analysis on research domains involving ultrasound, cancer management, pain, and other challenges (chemotherapy, surgical guidance, and postoperative care), to observe the trend by which the research field has grown over the years and propose a possible future trend. The data was obtained from the Web of Science, processed, and exported as plain text files for analysis in the Bibliometrix R web interface using the Biblioshiny package. A total of 3248 documents were identified from 1100 journal sources. A total of 390 articles were published in 2022, with almost a 100% growth rate from previous years. Based on the various network analysis, we conclude that the outcome of the constant research in this domain will result in better patient care during the management of various diseases, including cancer and other co-morbidities.

## 1. Introduction

Ultrasound (US) is an integral part of the clinical setup and plays a key role in many diagnostic procedures for cardiovascular conditions, cancers, neurological disorders, etc. US is successful in drug delivery and helpful in guiding certain complex surgeries. Several upgrades of the instrumentation and advanced methodologies involving the principle of ultrasound and acoustic waves have helped the disease management processes to make them easier and more effective in the clinical setup [1,2]. Ultrasound-induced neuromodulation is a growing methodology in the field of neuroscience, and ultrasound-mediated disease mitigation by stimulation of neuronal activity has gained the interest of experts recently [3]. Cancer presents several challenging comorbidities and needs significant attention from healthcare providers [4]. There are several studies on ultrasound with different methodologies that ventured its applications in the management of cancer at a diagnostic level where they aided other methodologies or directly helped in diagnosis. At the therapeutic level, US was reported for its ability in surgical guidance, drug delivery or sonodynamic methods, and palliative care, especially for pain management and for improvement of the physiological and psychological well-being of the patients [1,2,3,5]. 

Ultrasound-based therapy relies on high-intensity acoustic waves that stimulate the somatosensory nerves and modulates the pathogenesis of secondary complications of cancer with minimal invasiveness. Focused ultrasound was reported for its effectiveness against cancer-induced pain by ablation of peripheral nerves reversibly or irreversibly and was found to be more selective to C fibers than A fibers. Transient or irreversible blockage of nerve conduction was observed after High-intensity focused ultrasound (HIFU) treatment which was histologically demonstrated as axonal demyelination and Schwann cells injury. Studies have shown the positive effects of this biochemical neuromodulation in managing pain for patients with a wide range of severities of pain [5,6]. HIFU was utilized in several clinical studies involving the management of various complications during tumor/metastases conditions. The major success of HIFU was observed in the management of bone metastasis-induced pain by neuromodulation at the PNS level [7,8]. Reduction of pain sensation was reported after 3–14 days, and pain medication can be discontinued after the relief. Previously, pain due to bone metastasis or primary bone malignancies was treated with radiotherapy as a first-line treatment, followed by HIFU to mitigate the recurring pain [9]. HIFU is predominantly used for transient pains that could be resolved within a few days, thus giving patients faster relief from pain. There is no US dosage limitation, and repeated exposure can also be done for effective pain relief. Further, Magnetic Resonance (MR)-guided HIFU for pain relief was observed to be 52% cheaper than radiotherapy. Currently, several lesion-bearing bones like the leg, arm, scapula, sternum, sacrum, ribs, joints, etc., were successfully treated with HIFU and helped in a better quality of life. On the other hand, drawbacks of HIFU include lack of treatment plans for individual patients, the chance of histotripsy, thermal ablation, and patient not being eligible for US-based treatment due to other complications. At the Central nervous system (CNS) level, clinical trials on HIFU-mediated medial or central lateral thalamotomies for neuropathic pain management were successfully conducted with long-term efficacies. Neuromodulation at the Peripheral nervous system (PNS) has given great success for the US-based management of cancer-induced pain [8,10].

Cancer-mediated pain is one of the challenging symptoms to mitigate during the management of the actual disease because the stimuli are very strong due to the rapid growth of tumors and damage caused by the adjacent tissues. This directly increases the pain and, in turn, triggers the release of various inflammatory cytokines that can mediate the non-nociceptive pathway for pain conduction [7]. More than 60% of cancer patients suffer from extreme pain, and the World Health Organization (WHO) has developed guidelines for cancer pain management. However, the strategies to reduce cancer-mediated pain are still under par, although a significant number of research studies are recently focusing on addressing it [11,12,13]. Management of cancer-induced pain is currently dependent on non-opioids during mild pain; weak opioids are given to patients with moderate pain. At the same time, strong opioids, separately or in combination with non-opioids, are administered for patients presenting with severe pain due to cancer [14]. Drug-mediated pain relief and invasive neurostimulation with electrical or magnetic pulses have been showing significant success in pain management processes. However, the challenges posed by these strategies have made clinicians restrict their usage on a routine basis [3,5]. Sometimes, combinations of different management strategies were employed to obtain optimal pain relief for the patients [15]. Moreover, pain medications sometimes have a negative effect on the chemotherapeutic agents that are involved in the treatment of cancer progression [16]. Among the alternative therapies like diet-induced, behavioral therapy, etc., neurostimulation and modulation by ultrasound have gained the attention of clinicians in the management of cancer-induced pain [4]. Neurostimulation therapy was one of the alternative therapies that involved electrical, magnetic, or ultrasound-based stimulation [3,5].

There have been several studies in the past decade that demonstrated the effect of US and related methods of neurostimulation could be one of the reliable strategies for pain management [17,18]. However, the collaboration of US in the management of cancer and related pain conditions has been in existence since the late 1980s. However, the potential of US in the field of cancer and pain management was perceived from the late 1990s and early 2000s. In this study, we discuss the advances in the management strategies of cancer-mediated pain through ultrasound-guided therapy, followed by demonstrating the scientometric analyses of the ultrasound and its role in the mitigation of challenges and complications observed during cancer management. We have shown the annual production of scientific content (journal publications), and the network of keyword co-occurrence, collaboration, and co-citation were analyzed. Further, we have discussed the major conclusion drawn from some of the top-cited articles reporting the importance of ultrasound in cancer management.

### Objectives of the Bibliometric Analysis

Bibliometric analyses are considered a method to demonstrate the time-based progress observed in a research domain with respect to its multidisciplinary value. It is a quantitative analysis of the literature that indicates the annual scientific production and the most cited documents that provides a yearly trend of the topic in question [19]. Further, the role of the top contributing institutes and the top-cited articles gives more relevant research studies in the current scenario and patterns of collaborations between authors from different countries and institutes [20]. With constantly growing research in ultrasound, cancer, and the management cancer- induced complications, a significant evolution was observed in the respective fields and showed great prospects for multidisciplinary collaborations between different research groups that excel in their respective areas. Ultrasound has a long-standing relationship with biomedical imaging of many diseases, including cancer [2], while diagnosis and therapy of cancer and pertaining adverse conditions depend on biomedical instrumentations, including ultrasound. A potential for an interdisciplinary application was observed between cancer and ultrasound, either for diagnosis or therapy [21]. Researchers have considered this domain for exploration since the late 1990s, while it has been sporadically addressed since 1987. This bibliometric analysis has aimed at exploring the research trend that addressed the interdisciplinary approach to managing cancer and related complications using ultrasound. To arrive at our goal of observing the interdisciplinary potential of ultrasound in cancer and pain management, we designed the research questions as follows: 

RQ1: What is the diagnostic and therapeutic value of ultrasound in the management of cancer and related complications?

RQ2: How could research outcomes in ultrasound-mediated cancer management processes influence the areas of research that focus on ultrasound as a potential therapeutic tool for other complications?

Radiology and nuclear medicine showed their immense contribution to the timely and specified diagnosis of complex diseases, and at the same time, they were instrumental in guiding surgeries and targeted therapy with improved efficacy [22]. Cancer is a complex disease and requires support from imaging techniques such as ultrasound for clinically relevant diagnosis and optimal therapy [1]. In this regard, there is much scope for ultrasound and related imaging techniques for the diagnosis and treatment of cancer and related complications. Our bibliometric analysis was designed to provide a comprehensive view of the convergence between ultrasound and cancer-induced pain treatment. We aim to provide a trend in publications and citations related to the specific domain, followed by network analyses to give the thematic clusters of the specific research domain using 3248 published documents. These scientometric analyses help us with basic knowledge of the research domain and the collection of all data with their advances and inferences. Hence, the expected outcome of the bibliometric data is to arrive at a theme or domain which can bridge the gap between cancer, pain, and ultrasound. In the following sections, we have discussed the strategy for designing the bibliometric analysis, followed by an analysis of the results and then a discussion of the obtained analyses.

## 2. Research Methodology

Statistical analyses of the selected research domain were performed in three stages as previously described [23], and the methodology includes:

Stage 1: Data collection

Stage 2: Data processing

Stage 3: Data analysis

Data collection includes obtaining bibliometric data from the abstracting and indexing database. We used the Web of Science database maintained by Thomson Reuters. The rationale behind choosing the Web of Science database is that the quality of the obtained data is much better as they include publications that are indexed only by International Scientific Indexing (ISI) with information on the impact factors of the sources. The metadata obtained supports the analyses of the current scientometric data. The strategy to select a set of keywords was based on previous literature discussed in the previous sections in detail. Based on the keywords, three search strings were generated, one each for ultrasound, pain, and cancer (Table 1). Each search string consisted of similar terminologies; a search was performed with a combination of ultrasound or acoustics, while the search for cancer was combined with tumor as an alternative keyword. There is no prominent alternative keyword that denotes pain; hence we performed this search string as it is. We limited our search results to publications available in the English language that includes journal articles, book chapters, and proceedings. The inclusion of book chapters helps in providing the theory behind every advancement observed in the research domain, and proceedings papers give an idea about proof of concepts and novel ideas that are shared at the scientific forum. The obtained data was processed and exported as plain text files for analysis in the Bibliometrix R web interface using the Biblioshiny package. The information on the obtained data is briefed in Table 2. A total of 3248 results from Web of Science were obtained from 1100 relevant sources over a period of 35 years when the search was performed in combination with all three strings (“Ultrasound” or “Acoustics”, “Pain”, “Cancer” or “Tumor”). Then, the processed data was analyzed using R studio with the Biblioshiny package, which includes overall publication dynamics, network analysis, and systemic literature analysis of the top 20 articles.

## 3. Results

A bibliometric analysis provides the trend of a research domain in terms of research publications and citations. This gives the relevance of the research domain in the current scenario, and the information on highly cited articles gives an idea of the current focus of the researchers. Apart from the publication statistics, the collaboration network between authors from different countries and the network of the keywords that shows prevalence in the research community directly indicates the developments and upgradation of research findings in the research field.

### 3.1. Overall Information on the Published Articles

Our strategy in data acquisition has shown there are almost 3248 documents published related to complications during cancer management and the influence of ultrasound (Table 1). A total of 19,417 authors were identified among the published documents, and 81 of them were single authors. We found that there are 7.12 authors per document, and 10.5% of co-authorships are of international collaboration. The search for documents among two of the three strings resulted in a significantly high number (Table 1). When combining ultrasound and pain in the search, the results showed 360,050 documents that represent more than half of the published documents (544,352) in the ultrasound-related research domain (544,352). This suggests a significant overlap between ultrasound and pain research. However, cancer, as a research domain, is a long-standing challenge for the experts who are continuously exploring better management. More than 4 million published documents have been identified since 1987, and the combination of pain and tumor resulted in 1,464,408 documents, which are one-third of the documents related to cancer research that shows the link between pain and the pathogenesis of cancer. For the search with cancer and ultrasound combined, we obtained up to 1.7 million documents, which is evidence of the vital role played by ultrasound in various stages of cancer management over the years. The analysis was performed for the documents published between 1987 and 2022 (35 years) that have a very slight overlap between the three strings.

### 3.2. Annual Scientific Production

The total number of publications provides the relevance of the domain over the years since 1987. It was observed that there had been a consistent increase in the number of articles published over the years, and since 2017 there was an increase of at least 30 documents. In the years 2018 (217) and 2019 (274), there was an increase of up to 50 documents from the previous years. In the past decade, the number of publications has doubled, and there are already 748 publications published in the years 2021 and 2022, which was half the number of publications in the past decade (2011–2020). The annual growth rate was highest in the year 1997 (177%) when the number of publications increased to 32 from 18 in the previous year, but the growth rate in the past decade was considered significant as there was a consistent increase in the number articles and it was a minimum of 100% of the previous year (Figure 1).

### 3.3. Countries with Highest Publications and Citations

The research on ultrasound and its role in cancer management has attracted many research groups across the globe, as was evident from the number of publications produced by many developing and developed countries (Figure 2a,b). The USA topped the list with 2229 articles since 1987, followed by China with 1875 publications, and they top the list with the number of citations (USA—21,923; China—6047). South Korea remained in eighth place with 325 publications over 35 years, while the number of citations was about 2058, with 21.4 as average article citations. All the countries in the top 20 showed a minimum of 100 publications, except Poland, with 92 publications. The best average article citation was observed for Finland with 89.2, while Belgium and Austria had the second most average article citation with approximately 42 citations though Belgium has produced only 111 articles since 1987. However, Austria did not venture into the top 20 countries with the most scientific output in terms of publications, like Finland. The overall trend in the scientific contribution from each country looks very healthy, and the quality of the research was ensured by repeated citations.

### 3.4. Most Relevant Affiliation 

All the top 20 institutes ranked according to the number of publications were affiliated with at least a minimum of 30 articles (Figure 3). Zhejiang University from China published 81 articles, and the least of the top 20 was from the Carol Davila University of Medicine and Pharmacy, Romania, with 34 publications. Fudan University, China, and the University of Milan, Italy, published 38 publications, while 36 articles were published by three American institutes; the University of Texas MD Anderson Cancer Center, the University of Michigan, Harvard University, along with the Sapienza University of Rome. Ten out of the 20 institutes have published between 40–60 articles since 1987.

### 3.5. Sources with the Highest Production

Figure 4 shows the top 20 journal sources that published the most articles relevant to the research domain in question. Medicine journal has published 84 articles and is ranked the top journal, while the World Journal of Gastroenterology is in second place with 43 articles. Fifteen of the 20 journals have published 20–40 articles, and 18 articles were published in the Annals of Medicine and Surgery, which is seen at the 20th spot, while 19 articles were published each in Hepato-Gastroenterology and American Journal of Roentgenology. Ultrasound in Obstetrics & Gynecology, Journal of Pain Research, and Frontiers in Oncology have published 21 articles each related to cancer-induced pain and its management with ultrasound.

### 3.6. Publications with Highest Citations

The manuscript published by Ahmed and co-workers in *LANCET* in 2017 is a highly cited article. It is a comparative clinical study that involves the success of biopsy sampling guided by MRI and transrectal ultrasound (TRUS). The average citation per year is about 252.4, with a total of 1767 citations since publication. In comparison, the second most cited article was published by Peery and co-workers, who reviewed and updated the cost burden posed by GI, liver, and pancreatic diseases in the *Gastroenterology* journal in 2019. The average citation per year for this article is about 251.2 since its publication in 2019, and the total number of citations the article received is 1256. All the top 20 articles received a citation of more than 250, and the top 4 articles had citations of more than 500 (Table 3). Articles ranked from 6 to 14 received more than 300 citations, while an article published in 2014 by Singal and investigators in *PLoS Medicine* received 478 citations, with 47.8 citations per year for 8 years. The top two articles with the highest citation are a clinical trial and a clinical survey, respectively, and it clearly mentions the relevance and quality of the findings from these articles that could be applied in the clinical setup. In the current study, we found some of the highly cited articles did not directly focus on the role of pain in the pathological mechanisms of cancer or vice versa. Nevertheless, these papers do explore the impact of ultrasound on cancer and its management, including cancer-related pain, throughout various stages of cancer management processes like diagnosis of cancer, chemotherapy, post-operative care, etc. There are more than 1.7 million documents identified on ultrasound and cancer. This signifies the role of ultrasound in cancer management processes, and an obvious overlap of documents is inevitable even if we conduct a search that includes pain as the third research domain between ultrasound and cancer.

### 3.7. Word Cloud and Three-Fields Plot

The Sankey chart (Figure 5a) provides information on the research output that correlates the countries, authors, and keywords in the field of cancer-induced complications and ultrasound. China has published the maximum number of articles with 18 authors, namely Napoli A, Wang Y, Li J, Wang L, Li Y, Zhang Y, Zhang L, Liu Y, Catalano C, Wang W, Liang, P, Wang J, Yu, XL, Liu FY, Han ZY, Bhatnagar S, Cheng ZG, and Yu J. The USA comes in second place with seven authors. Then comes Australia at third place, and the least was observed from the UK with two authors. The most explored domains are cancer/tumor, ultrasonography, pain, surgery, diagnosis, complications, etc., and all 20 authors have conducted research in most of the keywords analyzed in our study. In the word cloud analysis (Figure 5b), keywords that dominated the bibliometric analysis were cancer, pain, diagnosis, ultrasound, tumor, and surgery which are the predominant keywords that denote the research outputs in the field of cancer management with ultrasound.

### 3.8. Network Analysis

Keyword co-occurrence analysis provides the uniqueness of each concept and its application in the research domain of interest [43]. Co-citation is an instance where the link between two different articles is established when a third document cites both articles. Apart from demonstrating the most cited papers, co-citation network analysis defines the starting point of the science towards its application for technology and is essential for the emergence of technology [44]. Further, collaboration networks provide information on authors from different countries working together for better development of technology. The increase in the number of collaborating countries indicates an improvement in the current research and technology [45].

#### 3.8.1. Co-Occurrence Analysis of Keywords

A Louvain clustering algorithm was utilized for the keyword analysis, and it considered authors’ keywords to form a network with 52 nodes/labels (Figure 6). The identified nodes were classified into three macro clusters that have nodes that are linked within the clusters and between the clusters, which clearly indicates the relevance and the amount of research work that has gone into the domain which we are studying. 

Cluster 1: Red cluster with 20 keywords is mostly related to cancer, management, and diagnosis, which are the most searched and studied keywords in the field of cancer management. Along with that, ultrasound, ultrasonography, biopsy, and computed tomography are some of the keywords that appeared in the red cluster.

Cluster 2: This cluster predominantly consists of therapeutic methods in the cancer management process, such as surgery, chemotherapy, radiofrequency ablation, radiation therapy, resection, etc. There are 16 keywords in this cluster, depicted in green consisting of certain cancer types that has many complications and are frequently studied at the preclinical level, such as adenocarcinoma, breast cancer, hepatocellular carcinoma, the safety of the therapeutic methods, etc.

Cluster 3: This cluster, depicted in blue color, mainly discusses the complications of cancer, such as pain, poor quality of life, morbidity, and clinical and pre-clinical studies on the effect of analgesics, anesthesia, and other cancer-induced complications. There are about 14 keywords observed in this cluster.

#### 3.8.2. Co-Citation Network

Co-citation network analysis resulted in four different clusters that contained manuscripts that deal with every aspect of the research domain (Figure 7). A total of 48 documents were identified, where 13 documents were identified in the red cluster, which contained two prominent articles published by Dr. Wiersema and coworkers, Dr. Gunaratnam and Dr. Aruna Sarma, affiliated with Mayo Clinic, Rochester, Minnesota, and St. Vincent Hospitals and Health Care Center, Indianapolis, Indiana. They reported a case study where endoscopic ultrasound was performed for a patient who was suffering from abdominal pain due to pancreatitis with pseudocysts, and the patient was successfully alleviated from the cysts and pain using endoscopic ultrasound. This study aided in a prospective study reported by the same group in 2001 headed by Wiersema, where 58 patients suffering from pancreatic cancer-induced pain were treated with endoscopic ultrasound-guided celiac plexus neurolysis (CPN). Both studies established that the endoscopic ultrasound technique was safer to perform, and in the prospective clinical trial, the conclusion was that CPN was safely performed with EUS [42,46]. EUS reported as a successful strategy in the mitigation of cancer pain, and additionally, the opioid side effects were avoided. EUS strategy was adapted from their previous results while targeting CPN for pain relief was approached with evidence provided by Eisenberg et al., who published a meta-analysis stating how CPN neurolysis can significantly help in cancer pain management [36]. The review was concluded with a recommendation that the CPN can be an answer for pancreatic or other intraabdominal cancer-induced pain with minimal adverse effects, and randomized controlled trials were recommended to obtain strong evidence for EUS-mediated CPN as an alternative to opioids.

The purple cluster consisted of 15 papers that predominantly deal with biopsy sampling with the guidance of US (mostly transrectal). Desgrandchamps and coworkers have developed a 3 Trosar technique for transperitoneal nephrectomy [47], which was developed on the basis of the transrectal biopsy sampling methods performed by authors such as Collin et al. in 1993 [48], Nash et al. in 1996 [49], and Clements et al. in 1993 [50]. All three studies involved prostate biopsy sampling through a transrectal route guided by ultrasound in which Collins, McKelvie, and co-workers performed the procedure and reported minimal complication and pain during the process [48], while Clements, Peeling, and co-workers reported no significant pain [50]. On the other hand, Nash, Shinhora, and co-workers reported reduced pain during prostate biopsy sampling with prior nerve blockade carried out by the transrectal ultrasound [49]. These studies mainly showed the ability of ultrasound in pain reduction during biopsy sampling-based cancer diagnosis. These clusters depicted the technological and clinical advancement in the research domain by utilizing ultrasound in the cancer diagnosis process-induced complications. 

Publications that appeared in blue and green clusters mainly dealt with cancer-induced pain mitigation. In the case of the blue cluster, the articles published by Catane and co-workers performed a preliminary study with focused ultrasound for palliation of pain in patients with bone metastases [51]. It was observed to be a collaborative study between, Catane, Gianfelice, and Liberman in 2007, 2008, and 2009 that provided valid evidence for ultrasound-mediated pain reduction in bone cancer patients, which is the three prominent articles in the blue cluster [52,53,54]. On the other hand, articles in green clusters deal with the reduction of pain during and post-surgery for breast cancer through nerve blockade at the thoracic wall using ultrasound. Blanco and co-workers showed the effect of ultrasound on thoracic nerve block at the serratus plane in providing pain relief to the patients [55], while Bashandy et al. and Kulhari et al. performed the pectoral nerve blockade during or after breast cancer surgery [56,57]. The study performed by Kulhari and co-workers from India compared and reported that the efficacy of pectoral nerve block using focused ultrasound was better than paravertebral block in postoperative pain relief [57].

The major observation of this co-citation network analysis is that the red and purple clusters which have the most number of articles, and they are published in the 1990s and early 2000s. These articles remained as the basis for the studies conducted in the past two decades involving ultrasound-mediated pain relief, which are in the other two clusters; apart from that, the articles published in the 1990s were cited continuously in the articles published in the past two decades in the respective cluster. This was demonstrated in the Historiographic direct citation network (Figure 8), which is an achronological citation network that develops the intellectual structure. Evidence provided by Dr. Wiersema and co-workers and Dr. Eisenberg and co-workers was the basis for many developments in the research domain. Both studies have established the use of endoscopic ultrasound for relieving abdominal pain due to cysts or metastatic tumors [36,42]. This was studied and cited by many future reviews and research that involves cancer pain relief, like Dr. Puli et al., 2009, Wyse et al., 2011, Kaufman et al., 2010, and Levy et al., 2008, who reported a systemic review on EUS-guided CPN for pancreatic cancer and pancreatitis. Similarly, the transrectal prostate biopsy was carried out comfortably with the help of ultrasound by Issa et al., 2000 Zisman et al., 2001 and Leibovici et al., 2002 [58,59,60]. High-intensity focused ultrasound was directly applied on the solid tumor for the treatment of pancreatic cancer by Wu et al. in 2005, and it was adapted by Wang et al. in 2011. These sets of studies have studied the mechanical or thermal effect of ultrasound on solid tumors, and these studies have been the steppingstone of the current research with HIFU as a potential treatment alternative for cancer that can be utilized individually in combination with other therapeutic modalities [61,62]. As mentioned before, the studies carried out by Catane et al., Gianfelice et al., and Liberman et al., have influenced several researchers to focus on ultrasound for better management of breast surgery-induced pain by blocking the pectoral nerve in the thoracic wall instead of paravertebral nerve blockade. This approach was successfully carried out by researchers such as Napoli et al. and Hurwitz et al., and they provided solid evidence for ultrasound-mediated nerve blockage and pain relief [51,52,54].

#### 3.8.3. Collaboration Network 

The Louvain cluster algorithm was adopted for the generation of a collaboration network between countries and pertaining affiliations that were involved in studies falling under the domain of ultrasound and challenges during cancer management. The collaboration network is depicted in Figure 9, where there are 3 clusters identified, and a total of 47 countries were observed to be collaborated for studying the effect of ultrasound on cancer, pain, or cancer-induced pain. With the greatest number of countries, the red cluster includes countries such as the USA, China, Japan, France, Korea, Australia, India etc. On the other hand, the blue cluster contained 11 countries, and the green cluster included 16 countries. In both of these clusters, the number of articles published is not on par with the USA or China. The clusters were differentiated based on the closeness to the research domain and the extent of collaboration for identifying and improvising the current technology. The green cluster includes countries like Germany, Netherlands, Denmark, Israel, etc., while the blue cluster includes the United Kingdom, Greece, Belgium, Portugal and etc. It was observed that all the countries have significant collaboration with other countries within and between the clusters, which denoted that researchers are open to interdisciplinary collaboration with other research groups across the globe for a better outcome.

## 4. Discussion

Cancer is one of the pathological stimuli that can inflict serious pain on patients, predominantly neuropathic pain, and sometimes can be a mixed type of pain with multiple etiology. Pain is one of the serious comorbidities of cancer that has a serious effect on the patient’s quality of life and can biologically interfere with the disease management process to a great extent [18]. Continuous development in cancer management has been reported for several decades, including cancer-induced complications [14]. However, mild to severe adverse effects are reported inconsistently with most of the drugs and are very risky to be administered for patients who are already suffering from other ailments [16]. Hence, physical methods like ultrasound, magnetic or electrical impulse-based stimulation, and neuromodulation have gained significant interest [3]. Among the physical methods, ultrasound has a clear edge over the others in terms of non-invasiveness and efficacy in deeper tissues [5]. Hence, in this study, we discussed about the issues faced during cancer management and recent developments in ultrasound-mediated cancer management. After a discussion of the current literature, we analyzed the trends in publications, citations, and collaboration that have been a great influence on the developments observed in the field of ultrasound-mediated cancer management.

### 4.1. Research Implications

The bibliometrics analysis performed in the current study has given us the prospects of ultrasound in the management of complications during cancer. The convergence of three different research domains (ultrasound, cancer, and cancer-induced complications) was clearly understood from this study. The initial research focus, according to the analysis, was on ultrasound in enabling cancer management, especially during the diagnostic stages [42,46]. Continuous research studies observed a trend over the years on ultrasound and its clinical implications; the focus shifted to the management of cancer in therapeutic stages. On the other hand, the physiological or biochemical modulation during ultrasound application has aided the expert in exploring the possibility of ultrasound in pain management [51,52]. Developments observed in the field of ultrasound had some serious positive influence on the management of chronic pain like rheumatoid arthritis, lower back pain, migraine, etc. [5,63,64,65,66]. This has paved the way for utilizing the effect of ultrasound on pain reduction during any of the cancer management processes like diagnosis, therapy, and post-operative care [48,49,56,57]. Biopsy-induced pain was managed significantly using transrectal ultrasound, and later ultrasound was used in the management of breast cancer-related surgery-induced pain, and since then, it has been a part of palliative care after cancer treatment methods like chemotherapy, surgery, or radiotherapy, that could trigger the pain pathway [56,57]. The role of ultrasound in pain management during the progression of bone cancer/metastases and after treatment was invaluable [52,53]. Ultrasound has been instrumental in reducing the burden faced by the hospital sector due to increasing pain incidences and also an improvement in the patient’s quality of life, better sleep, and daily activities [67,68]. Recent studies were focused on utilizing the deep penetrative ability and non-invasiveness of ultrasound in neuromodulation through spinal nerve stimulation for the management of various types of pain, especially cancer-induced pain with focused ultrasound predominantly at higher intensity [3].

Analyses performed in the current bibliometric analysis have given us an overview of the research trends and evolution in cancer management. Further, the study provided evidence of the emergence of ultrasound as a successful tool for helping researchers and clinicians with a better treatment process without pain symptoms. The convergence of research trends from all three domains (ultrasound, pain, and cancer) was clearly observed from our network analysis, and the relevance of the research domain among researchers across the world was observed with an increasing number of publications and collaboration network analysis. 

### 4.2. Future Implications

The initial success of ultrasound in any disease management process was the non-invasiveness and better spatiotemporal resolution with deeper tissue penetration compared to other imaging or therapeutic technologies [69]. In the case of pain management due to cancer or any other ailments, the influence of ultrasound was significant and is considered the better alternative to analgesic medications, behavioral treatments, or other types of physical neurostimulation (magnetic or electrical) [8]. Based on our bibliometric analyses, the future of cancer management depends on ultrasound and its developments. In this regard, the hand-held ultrasound device for cancer pain relief at the patient’s convenience can be developed and further reduce hospital visits and medications. Improvements in the ultrasound instrumentation must be made so that easy handling for the attendee or bystander in the palliative care, similar to other routine devices like a thermometer, digital sphygmomanometer for blood pressure monitoring, arm patch for vitals monitoring, and glucometer for blood sugar level monitoring and etc., will reduce the routine or continuous patient monitoring for observing the complications. A combinatorial approach with medications and ultrasound or sonodynamic therapy that involves activation of drugs using ultrasound application. Applications of nano/micro formulations and bioactive materials made of anti-cancer drugs that are responsive to ultrasound can influence the reduction in cancer-induced complications with fewer adverse effects due to drugs. 

## 5. Conclusions

The study has discussed in detail cancer management strategies, with a special focus on cancer-induced pain. We conducted our bibliometric analyses based on two major research questions:

RQ1: What is the diagnostic and therapeutic value of ultrasound in the management of cancer and related complications?

The network analyses performed in our bibliometric study recognized specific clusters where ultrasound and cancer-related pain management are cited together in a single document. With an increase in the usage of ultrasound in the management of many types of pain, these research works give special regard to ultrasound among many strategies for reducing cancer pain. Companies involved in the production of ultrasound instruments and their R & D sector will be keen to develop ultrasound instrumentation that is portable and suitable for on-demand application for pain. This will reduce the cost required for critical care and the necessity of a skilled technician to perform the ultrasound application.

RQ2: How research outcomes in the ultrasound-mediated cancer management process could influence the areas of research that focus on ultrasound as a potential therapeutic tool for other complications?

Major research trends were focused on ultrasound-mediated management strategies during bone metastasis therapy, breast cancer surgery, and reduction of pain during tissue biopsy sampling from prostate cancer patients. It is a very intriguing field of study, and with the growth that was observed in the current bibliometric analysis, the use of ultrasound in every stage of the cancer management process is inevitable. The use of ultrasound in Rheumatoid arthritis, gouty arthritis, urinary stone-induced pain (during and after treatment), diabetic neuropathy-induced pain complications, Parkinsonism, spinal cord-related conditions, etc.

Although the study identifies positive trends in the research domain with improved scientific production and global collaboration, this bibliometric analysis has some drawbacks, like selecting the documents only from the Web of Science, usage of generalized keywords like pain and other cancer-related management, which will be addressed in the further studies by involving Scopus, PubMed, Cochrane and etc., for documents search and specific keywords like bone cancer, breast cancer and other types of cancer along with certain keywords like pain, inflammation, postoperative care and monitoring and others while searching for documents related to the management of cancer-induced complications. The study will be followed by a systematic review with meta-analyses and deep machine-learning approaches for understanding the current research trend of the selected domain. We planned to study the technological forecast to predict future trends in convergence between the research fields.

## Figures and Tables

**Figure 1 sensors-23-07290-f001:**
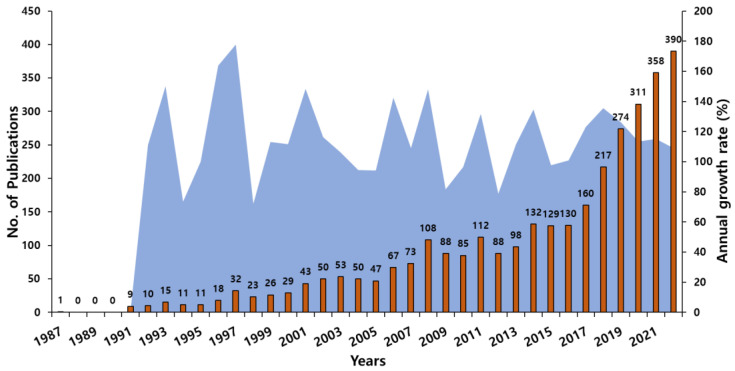
Annual scientific production and its growth rate.

**Figure 2 sensors-23-07290-f002:**
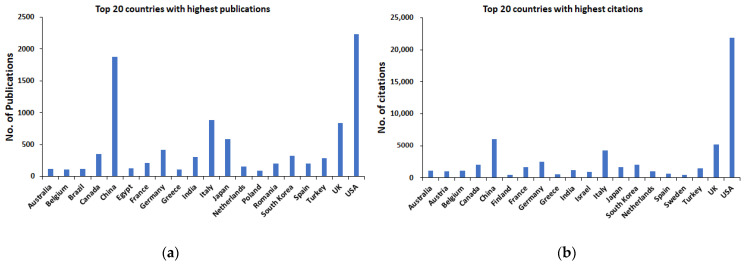
Top 20 countries with highest publications (**a**) and citations (**b**).

**Figure 3 sensors-23-07290-f003:**
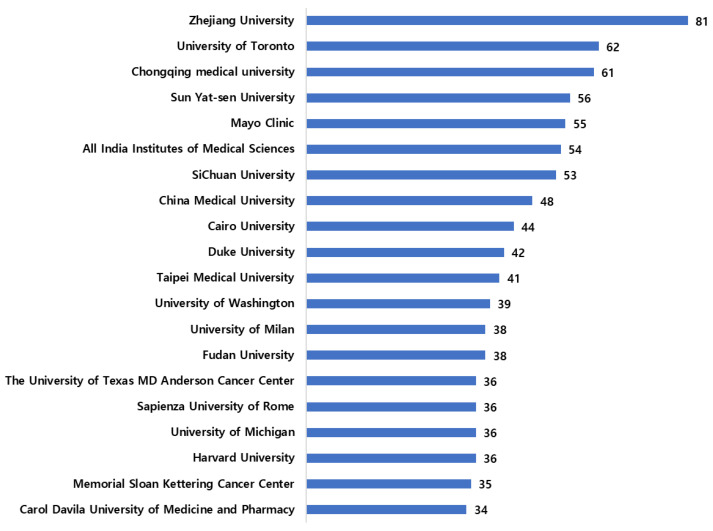
Top 20 institutes based on article count.

**Figure 4 sensors-23-07290-f004:**
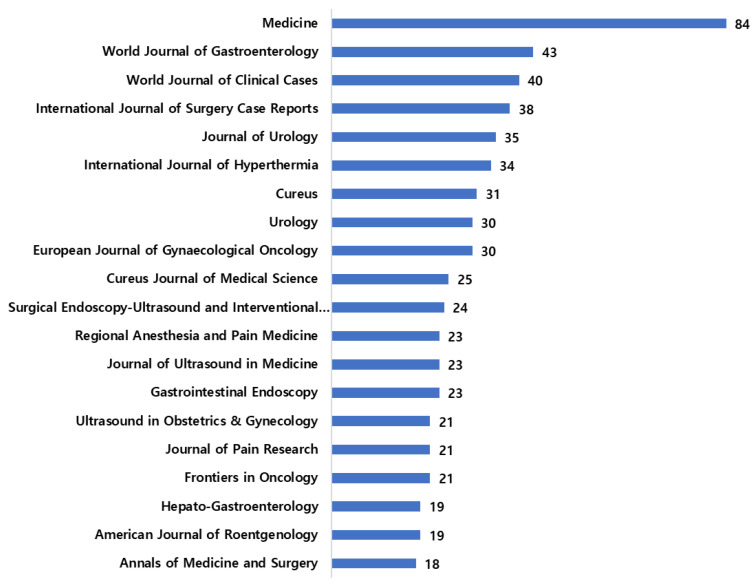
Top 20 relevant sources.

**Figure 5 sensors-23-07290-f005:**
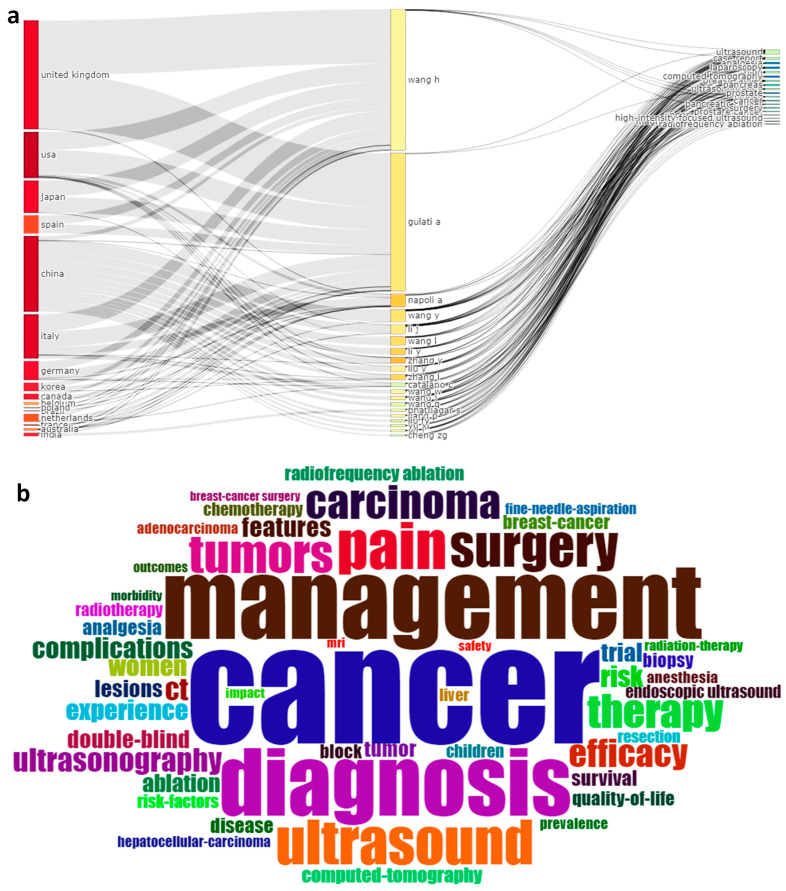
Three-field plot (**a**) and Word cloud (**b**).

**Figure 6 sensors-23-07290-f006:**
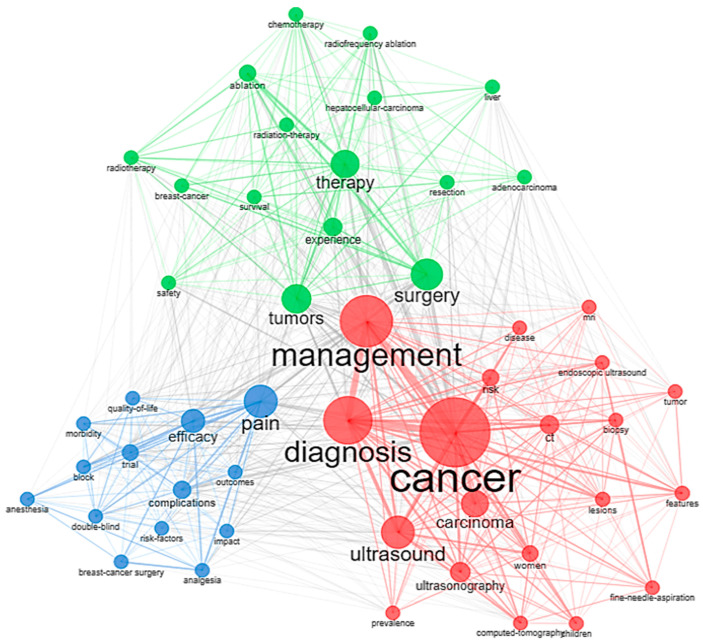
Co-occurrence of keywords.

**Figure 7 sensors-23-07290-f007:**
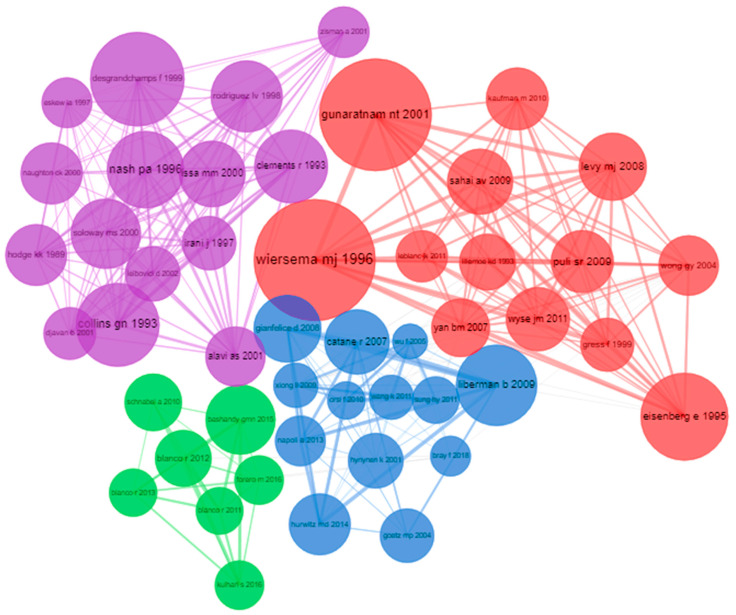
Co-citation network.

**Figure 8 sensors-23-07290-f008:**
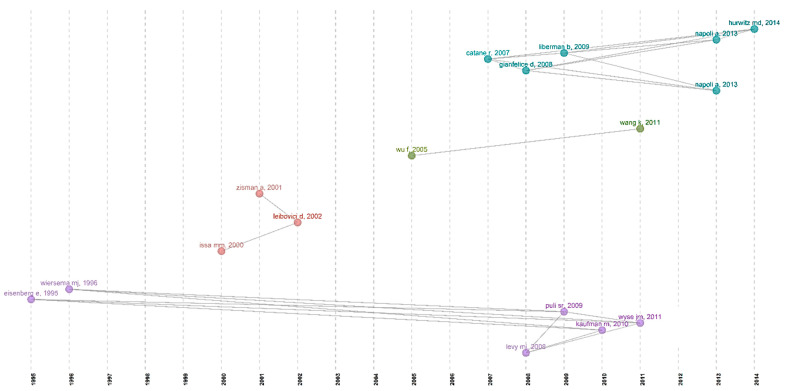
Historiographical direct citation network.

**Figure 9 sensors-23-07290-f009:**
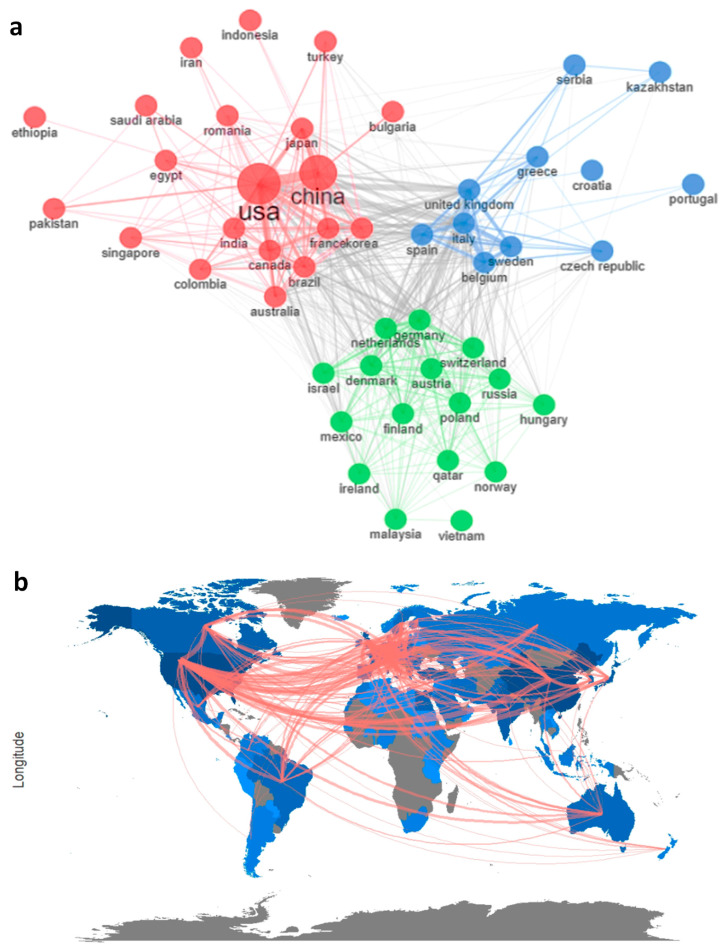
Collaboration network (**a**) and collaboration map (**b**).

**Table 1 sensors-23-07290-t001:** Search Strategy.

S. No	Topic	Results	Combination
1	“Ultrasound” or “Acoustics”	544,352	#1 and #2 and #3—3248 #1 and #2—360,050 #2 and #3—1,464,408 #1 and #3—1,737,421
2	“Pain”	814,875	
3	“Cancer” or “Tumor”	4,157,876	

**Table 2 sensors-23-07290-t002:** Main information about data.

Description	Results
Timespan	1987:2022
Sources	1100
Documents	3248
Average citations per document	19.2
Keywords Plus (ID)	5169
Author’s Keywords (DE)	8345
Authors	19,417
Authors of single-authored documents	81
Single-authored documents	83
Co-authors per document	7.12
International co-authorships %	10.53

**Table 3 sensors-23-07290-t003:** Top 20 globally cited publications.

Rank	Title	Author and Source	Year of Publication	Total Citations	Total Citations per Year	Reference
1	Diagnostic accuracy of multi-parametric MRI and TRUS biopsy in prostate cancer (PROMIS): a paired validating confirmatory study	Ahmed HU, *LANCET*	2017	1767	252.43	[24]
2	Burden and cost of gastrointestinal, liver, and pancreatic diseases in the United States: Update 2018	Peery AF, *Gastroenterology*	2019	1256	251.20	[25]
3	Systematic review of complications of prostate biopsy	Loeb S, *European Urology*	2013	650	59.09	[26]
4	High-intensity focused ultrasound: surgery of the future?	Kennedy JE, *The British Journal of Radiology*	2003	511	24.33	[27]
5	Early detection, curative treatment, and survival rates for hepatocellular carcinoma surveillance in patients with cirrhosis: A meta-analysis	Singal AG, *Plos Medicine*	2014	478	47.80	[28]
6	2015 Gout Classification Criteria	Neogi T, *Arthritis & Rheumatology*	2015	371	41.22	[29]
7	High-intensity focused ultrasound for the treatment of liver tumors	Kennedy JE, *Ultrasonics*	2004	343	17.15	[30]
8	Logistic regression model to distinguish between the benign and malignant adnexal mass before surgery: A multicenter study by the international ovarian tumor analysis group	Timmerman D, *Journal of Clinical Oncology*	2005	332	17.47	[31]
9	Ovarian carcinoma diagnosis: Results of a national ovarian cancer survey	Goff BA, *Cancer*	2000	323	13.46	[32]
10	Focused ultrasound treatment of uterine fibroid tumors: Safety and feasibility of a noninvasive thermoablative technique	Stewart EA, *American Journal of Obstetrics & Gynecology*	2003	314	14.95	[33]
11	Radiofrequency ablation of benign thyroid nodules: safety and imaging follow-up in 236 patients	Jeong WK, *European Radiology*	2008	312	19.50	[34]
12	EULAR recommendations for the use of imaging in the diagnosis and management of spondyloarthritis in clinical practice	Mandl P, *Annals of Rheumatic Diseases*	2015	308	34.22	[35]
13	Neurolytic celiac plexus block for treatment of cancer pain: A meta-analysis	Eisenberg E, *Anesthesia & Analgesia*	1995	306	10.55	[36]
14	2015 Gout classification criteria: an American College of Rheumatology/European League Against Rheumatism collaborative initiative	Neogi T, *Annals of Rheumatic Diseases*	2015	305	33.89	[29]
15	Screening for prostate cancer (Review)	Ilic D, *Cochrane Library: Cochrane Reviews*	2013	291	26.45	[37]
16	Value of endoscopic ultrasound-guided fine needle aspiration biopsy in the diagnosis of solid pancreatic masses	Voss M, *Gut*	2000	288	12.00	[38]
17	Continuous peripheral nerve blocks: A review of the published evidence	Ilfeld BM, *Anesthesia & Analgesia*	2011	284	21.85	[39]
18	Complication rates and risk factors of 5802 transrectal ultrasound-guided sextant biopsies of the prostate within a population-based screening program	Raaijmakers R, *Urology*	2002	279	12.68	[40]
19	Mapping the human genetic architecture of COVID-19	Niemi MEK, *Nature*	2021	278	92.67	[41]
20	Endosonography-guided cystoduodenostomy with a therapeutic ultrasound endoscope	Wiersema MJ, *Gastrointestinal Endoscopy*	1996	270	9.64	[42]

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
