# Peer review of "The Role of Ultrasound in Cancer and Cancer-Related Pain—A Bibliometric Analysis and Future Perspectives"

_sensors, 2023, doi:10.3390/s23167290_

Round 1

Reviewer 1 Report

Thank you for submitting your valuable manuscript to this journal.

This paper is a descriptive bibliometric analysis of publications in the field of ultrasound mediated cancer pain management. The aim of the study is to observe the trend by which the research domain has grown over the years. The research question is about therapeutic value of ultrasound in management of cancer related pain.

The authors found 3248 documents related to cancer, but unfortunately it seems that they did not read the papers to include the most relevant documents related to their study questions. For example, the most cited publication that was included in the study is a comparative clinical study, which is aimed to validate diagnostic accuracy of multi-parametric MRI and Trans-rectal ultrasound (TRUS) biopsy in prostate cancer. The second most cited paper evaluated the Burden and cost of gastrointestinal, liver, and pancreatic diseases in the United States. None of these studies are about pain management with ultrasound!

The English language is fine and only moderate editing is needed.

Author Response

The authors whole heartedly thank the reviewer for this constructive remark. 

Reviewer 2 Report

The article is well written and iteresting. good job

Author Response

Authors thank reviewer for the comment and appreciation

Reviewer 3 Report

High Intensity ultrasound issue in in pain reduction to the patients suffer from different type of cancer is an interest topic

Abstract is clear to me just add justification to your review article

Introduction is well structured and fantastic reading and soundness language.

The methodology of review article from web of science for more than 1100 journals is well concluded and author/s mentioned the important points and comments that are well related to the current review article

Discussion is acceptable in the current form.

English is fine and well revised.

Author Response

Authors thank reviewer for the comment and appreciation. Authors have made changes in the abstract as per reviewers’ suggestion 

Round 2

Reviewer 1 Report

Thank you for your response to the feedback.

As you have explained in the response, cancer management comprises many stages including diagnosis, treatments, prognostic evaluation, surgery and etc. Ultrasound has a role in many parts of cancer management and it is important to pay attention to this useful modality.

Since the aim of the present study is just on the therapeutic value of ultrasound in the management of cancer-related pain, it is essential to only include the documents on this specific topic. With a little review of the included papers, you can easily find out that many of the included documents in this work are irrelevant to pain management.

I would recommend you revise the title of the paper and remove the word pain. Please also change the aim of the study to the general applications of ultrasound in the management of cancer.

Good luck.

Moderate editing of English language required.

Author Response

Point 1: Thank you for your response to the feedback.

As you have explained in the response, cancer management comprises many stages including diagnosis, treatments, prognostic evaluation, surgery and etc. Ultrasound has a role in many parts of cancer management and it is important to pay attention to this useful modality.

Since the aim of the present study is just on the therapeutic value of ultrasound in the management of cancer-related pain, it is essential to only include the documents on this specific topic. With a little review of the included papers, you can easily find out that many of the included documents in this work are irrelevant to pain management.

I would recommend you revise the title of the paper and remove the word pain. Please also change the aim of the study to the general applications of ultrasound in the management of cancer.

Good luck.

Response 1:

Authors thank the reviewer for the comment and recommendation.

We have changed the title of the manuscript as given below:

Old title: “Ultrasound-Mediated Cancer Management processes: Biblio-metric Analysis and Future Perspectives”

New title: “The Role of Ultrasound in Cancer and Cancer-Related Pain - A Bibliometric Analysis and Future Perspectives”

Authors decided to retain the word “pain” title due to the significant involvement of “pain” during any of the cancer management processes and involvement of “ultrasound” in management of cancer and cancer related pain. The decision was likely based on the findings of the bibliometric search, which revealed that there is a considerable body of literature available related to the use of ultrasound in cancer treatment and the role of pain in various aspects of cancer management. By focusing on the topic of "ultrasound in cancer and cancer-related pain," the authors aim to shed light on the research trend regarding the role of ultrasound technology in diagnosing and treating cancer along with management of pain during the course of cancer treatment with ultrasound.

We modified the title as “The Role of Ultrasound in Cancer and Cancer-Related Pain……..” and required modifications were made in the abstract, introduction, study objectives (aim of the study) and conclusion sections to make the content suitable for the title. The introduction section was reorganized to focus on role of ultrasound as a pivotal strategy in cancer management or cancer related pain management.

Additionally, as a clarification, we have also added the number of documents obtained as a result of different combination of search performed with the 3 research strings and included that in Table 1.

Table 1. Search Strategy

S.No

Topic

Results

Combination

1

"Ultrasound" or "Acoustics"

544,352

#1 and #2 and #3 - 3248

#1 and #2 - 360,050

#2 and #3 – 1,464,408

#1 and #3 - 1,737,421

2

"Pain"

814,875

3

"Cancer" or "Tumor"

4,157,876

We have also discussed this content in the results section for giving an idea to the readers that ultrasound plays significant role in cancer management process including pain reduction. The content added in the manuscript at

Page 6, line 190 to 202:

“Search for documents among two of the 3 strings resulted in a significantly high number (Table 1). When combining ultrasound and pain in the search, the results showed 360,050 documents that represents more than half of the published documents (544,352) in the ultrasound-related research domain (544,352). This suggests a significant overlap between ultrasound and pain research. But, Cancer as a research domain, is a long-standing challenge for the experts who are in continuous exploration for better management. More than 4 million published documents were identified since 1987 and combination of pain and tumor resulted in 1,464,408 documents that are 1/3rd of the documents related to cancer research which shows the link between pain and pathogenesis of cancer. While search combined with cancer and ultrasound, we obtained up to 1.7 million documents, which is evidence for the vital role played by ultrasound in various stages of cancer management over the years. The analysis was performed for the documents published between 1987 to 2022 (35 years) that has a very slight overlap between the 3 strings.”

Page 8, line 267 to 275:

In the current study we found some of the highly cited articles, not directly focused on role of pain in pathological mechanisms of cancer or vice versa. Nevertheless, these papers do explore the impact of ultrasound on cancer and its management, including cancer-related pain, throughout various stages of cancer management processes like diagnosis of cancer, chemotherapy, post operative care, and etc. There are more than 1.7 million documents identified between ultrasound and cancer. This signifies the role of ultrasound in cancer management processes and an obvious overlap of documents is inevitable even if we conduct the search that include pain as the 3rd research domain between ultrasound and cancer.
